# Association between Participation in the Short Version of a Workplace Oral Health Promotion Program and Medical and Dental Care Expenditures in Japanese Workers: A Longitudinal Study

**DOI:** 10.3390/ijerph19053143

**Published:** 2022-03-07

**Authors:** Yuki Mochida, Shinya Fuchida, Tatsuo Yamamoto

**Affiliations:** 1Department of Dental Sociology, Kanagawa Dental University, Yokosuka 238-8580, Japan; mochida@kdu.ac.jp; 2Department of Education Planning, Kanagawa Dental University, Yokosuka 238-8580, Japan; fuchida@kdu.ac.jp

**Keywords:** oral health promotion program, dental care expenditure, medical care expenditure, number of days of dental treatment, number of days of medical treatment

## Abstract

Studies suggest that intensive oral health promotion programs in the workplace reduce dental and medical care expenditures. The purpose of this longitudinal study was to evaluate the short version of an oral health promotion program in the workplace from the viewpoint of dental and medical care expenditures. Data for annual expenditures and number of days of dental, periodontal, and medical treatment in fiscal year 2018 and participation in the short version of a workplace oral health promotion program of 2545 workers (20–68 years old) in a company in fiscal year 2017 and prior were obtained. Zero-inflated negative binomial regression models or negative binomial regression models were used to evaluate the association between participation in the program and expenditures or number of days of treatment after adjusting for sex and age. Program participants were more likely than non-participants to visit dentists for dental and periodontal treatment. Those who participated twice or more spent less on dental, periodontal, and medical treatment and had fewer visits to dentists than non-participants. These results suggest that the short version of an oral health promotion program in the workplace decreases expenditures for dental, periodontal, and medical treatment.

## 1. Introduction

The oral health status of Japanese children, especially the prevalence of dental caries, has improved over the last two decades; however, there has been insufficient improvement in the oral health status of adults [1]. The prevalence of dental caries and periodontal disease in middle-aged individuals has not changed in recent years [2]. Oral health promotion programs including oral health examinations and oral health instruction for children are well implemented by municipalities because programs for 1.5- and 3-year-olds and schoolchildren are mandatory in Japan. In contrast, oral health promotion programs are not compulsory for adults. Less than two-thirds of the municipalities in Japan have programs focused mainly on periodontal examination, and participation rates are lower than 5% [3].

Workplaces are suitable for screening of dental diseases and oral health instruction because most of the target population would be involved. Studies have shown that workplace oral health promotion programs improve the oral health status of employees [4,5] and contribute to saving costs associated with dental treatment [6]. However, few workplaces have such systems in Japan [7]. Some large companies have their own oral health promotion programs; however, small companies do not [8].

There is some resistance to introducing oral health promotion programs in the workplace from the management of such workplaces, from the dental profession, and from employees [9]. For example, the Community Periodontal Index (CPI) is often used for the screening of periodontal disease in public health [10]; however, qualified examiners are needed to perform the probing procedure. In addition, considerable time and effort are also required when screening for periodontal disease in a large number of subjects. In fact, oral health promotion programs in some previous studies have included intensive oral health instruction, and the total time required for each subject was approximately 20 min or longer [4,6,11]. Such time-consuming programs would not be introduced to the workplace because companies do not want to lose working hours to the implementation of such programs.

We have established a simplified and shortened oral health promotion program to facilitate introduction of the program to medical checkups in workplaces in Japan. Because almost all workplaces have medical checkup programs, it would be easy to introduce a short oral health promotion program into existing medical checkups. Our short oral health promotion program includes screening for periodontal disease using a questionnaire and a kit measuring the salivary lactate dehydrogenase level, screening for dental caries, and tooth brushing instruction. We have already confirmed the effectiveness of the instructional portion of the program, such as through improvement of periodontal disease, in our previous study [12].

An economic evaluation of an oral health promotion program is valuable for persuading the management of workplaces, i.e., employers and policy makers, to introduce oral health promotion programs in the workplace. Dental care expenditures and frequency of dental visits are useful for assessing the economic impact of an oral health promotion program. A universal healthcare system including most dental treatments was established in Japan in 1961, and every resident is enrolled in some form of health insurance plan in Japan. Employees of large companies are covered by a group health insurance plan managed by their employer, i.e., a health insurance association. Most dental care expenditures are covered by health insurance, excluding the cost of orthodontic treatment and part of the cost of any prosthetic appliance, such as a dental implant.

Because poor oral health is associated with chronic systemic diseases such as cardiovascular diseases, diabetes mellitus, and cancers [13], introduction of a workplace oral health promotion program may decrease medical costs. 

The purpose of this study was to compare dental and medical care expenditures and the frequency of dental and medical visits between participants and non-participants in the short version of an oral health promotion program in a workplace belonging to one health insurance association.

## 2. Materials and Methods

### 2.1. Study Design and Participants

A longitudinal study was conducted in a company (educational business) belonging to one health insurance association in fiscal year 2017–2018 (from April 2017 to March 2019). The inclusion criteria of subjects were all workers who were employed by the company during the whole study period. Exclusion criteria of the subjects were those who did not consent to this study, who did not have health insurance in fiscal year 2018, and who were not employed by the company when the oral health program was performed in fiscal year 2017.

This health insurance association recommended that the company introduce our short oral health program within their annual medical checkups, and the company has conducted the program since 2011. Participation in the program was voluntary; however, the receptionist of the medical checkup encouraged workers to participate. The purpose of the program is to motivate participants to adopt proper oral health behaviors to prevent periodontal disease and dental caries. Proper oral health behaviors include interdental brushing to prevent periodontal disease and use of fluoride toothpaste to prevent dental caries. Another purpose of the program is to encourage participants to visit dentists for treatment if they have suspected dental diseases. The oral health program consisted of screening for periodontal disease and dental caries and oral health instruction by dentists. Participants were expected to fill out a questionnaire for periodontal disease screening [14] in advance and bring it with them on the day of the program. They then underwent an additional screening test for periodontal disease using a rapid test kit [15,16], which included a rapid measurement of lactate dehydrogenase in saliva within one minute. The rapid measurement of lactate dehydrogenase in saliva did not require specific conditions. While waiting for test results, examination of the oral cavity, including for numbers of teeth present and decayed teeth, was performed by a dentist to give explanations to the participants regarding their dental health status and to recommend visiting a dentist if they had decayed teeth. Then, oral health instruction, such as person-to person tooth brushing instruction using the toothpick method [17], was provided on the basis of these results. The questionnaire included items on previous participation in the program; therefore, the dentists were able to use this information from the participants’ answers to tailor their oral health instruction. Time needed for the oral health program varied from three to five minutes per participant. When the participants had dental problems and wanted consultation from the dentist, the time spent tended to be longer, but no longer than ten minutes.

All participants were informed that their data would be used for research (opt-out system) and be de-identified. Information about sex and age was retrieved from the de-identified data. The protocol of this study was reviewed and approved by the Ethical Committee of Kanagawa Dental University (No. 620 and 689).

### 2.2. Measures

Data for annual medical expenditures and number of days of treatment per person from the health insurance claims in fiscal year 2018 were provided by the health insurance association. Medical expenditures were divided into dental care expenditures and medical care expenditures, including both inpatient and outpatient care. Similarly, the number of days of treatment was divided into those for dental treatment and those for medical treatment. Periodontal care expenditures, including those for periodontal examination, oral health guidance by a dental hygienist, scaling and root planing, pocket curettage, and periodontal treatment, were extracted from the dental care expenditures as the number of days of periodontal treatment and from those of dental treatment. Those with a diagnosis of diabetes mellitus and diabetes-related complications were categorized as having diabetes mellitus in order to assess the burden of treatment costs by disease. Then, the data for personal health insurance claims were linked to those of the presence or absence of participation in the workplace oral health program in fiscal year 2017 (from June 2017 to March 2018). In addition, data regarding previous participation in the program were also added to the database.

### 2.3. Statistical Analysis

Participants were categorized into three groups according to the number of times they had participated in the short version of a workplace oral health promotion program (never, once, and twice or more), and their information was obtained from the records of participation in the program and the completed questionnaires for the participants of the program in fiscal year 2017. The three groups were compared in terms of sex and diagnosis of diabetes mellitus using the Chi-squared test. Age, expenditures, and number of days of dental, periodontal, and medical treatment were compared among the three groups using the Kruskal–Wallis test followed by pairwise comparisons using the Mann–Whitney *U* test with the Bonferroni correction.

In order to allow for excess zeros in count variables such as expenditures and number of days of dental and periodontal treatment, zero-inflated negative binomial regression models [18] were used to evaluate the association between participation in our oral health promotion program and expenditures or number of days of treatment after adjusting for sex and age, both of which are associated with expenditures and number of days of dental treatment [19]. Zero-inflated negative binomial regression modeling generated two separate models and then combined them: first, a logit model was generated for zeros, predicting whether the expenditures or numbers of days of treatment were not counted; second, a negative binomial model was generated, predicting the counts for expenditures or number of days of treatment for subjects with >0 expenditures or number of days of treatment; and finally, the two models were combined. The expenditures and the number of days of medical treatment were estimated with negative binomial regression, adjusting for sex and age (Model 1). Then, diagnosis of diabetes mellitus was added to evaluate whether diabetes mellitus was a confounder in the association, because diabetes mellitus is associated with periodontal disease [20] (Model 2). The zero-inflated parts of zero-inflated negative binomial models were used to calculate the odds ratios (ORs) and their 95% confidence intervals (CIs). The estimates for the negative binomial parts of zero-inflated negative binomial regression models and negative binomial regression models are presented as rate ratios (RRs) with their 95% CIs. For the zero-inflated negative binomial regression models and the binomial regression models, analyses stratified by sex were conducted in addition to those using all subjects. All statistical analyses were performed using Stata/MP (version 16.0; Stata Corp. LLC, College Station, TX, USA).

## 3. Results

Subjects included in the analyses were a total of 2545 workers (1059 males and 1486 females; age 20–68 years, mean age 40.8 years, standard deviation 9.1 years) after excluding two subjects who did not consent to this study, 188 subjects who did not have health insurance in fiscal year 2018, and 263 subjects who were not employed by the company when the oral health program was performed in fiscal year 2017 (Figure 1).

The participation rate in the short oral health program in fiscal year 2017 was 50.6% (*n* = 1287), with 3.4% (*n* = 86) being first-time participants and 47.2% (*n* = 1201) being those participating for at least a second time (Table 1).

The comparison of each variable among non-participants, one-time participants, and those who had participated twice or more in the short oral health promotion program is shown in Table 1. Program participants were characterized by younger age. Twice-or-more participants visited medical doctors more frequently than non-participants. Almost half of all study subjects had a total number of zero for dental and periodontal care expenditures and for the number of days of dental and periodontal treatment.

The results of the zero-inflated negative binomial regression analyses using expenditures and the number of days of dental and periodontal treatment as outcome variables are shown in Table 2. For the zero-inflated part, the ORs (95% CI) of being “free of dental care expenditures” for one-time and twice-or-more participants were 0.47 (0.35–0.62) and 0.82 (0.64–1.05), respectively (reference: non-participants). The ORs (95% CI) for periodontal care expenditures for one-time and twice-or-more participants were 0.48 (0.35–0.67) and 0.79 (0.67–0.92), respectively (reference: non-participants). Of the subjects having >0 Japanese yen of dental and periodontal care expenses in the negative binomial model part, the RRs (95% CI) of the twice-or-more participants were 0.84 (0.83–0.85) and 0.85 (0.83–0.87), respectively (reference: non-participants). 

For the zero-inflated part, the ORs (95% CI) of being “free of dental treatment” for one-time and twice-or-more participants were 0.29 (0.15–0.56) and 0.67 (0.53–0.85), respectively (reference: non-participants) (Table 2). The ORs (95% CI) of being “free of periodontal treatment” for one-time and twice-or-more participants were 0.27 (0.14–0.53) and 0.65 (0.55–0.76), respectively (reference: non-participants). In the negative binomial model parts of the zero-inflated negative binomial model for the number of days of dental and periodontal treatment, the RRs (95%) for the twice-or-more participants were 0.79 (0.78–0.79) and 0.80 (0.74–0.87), respectively (reference: non-participants), in the subjects with ≥1 days of dental and periodontal treatment.

Table 3 shows the results of the zero-inflated negative binomial regression analyses for expenditures and the number of days of dental and periodontal treatment in males and females after adjusting for age. In males, the ORs of the zero-inflated part for one-time participants (reference: non-participants) were significantly lower. In females, the ORs of the zero-inflated part and the RRs of the negative binomial part for twice-or-more participants (reference: non-participants) were significantly lower, except the OR of the dental care expenditure. 

The RRs for medical care expenditures and the number of days of medical treatment using the negative binomial regression models are shown in Table 4. The RR (95% CI) of medical care expenditures for the twice-or-more participants was 0.81 (0.77–0.85) (reference: non-participants) in Model 1. In Model 2, the RR (95% CI) of medical care expenditures for the twice-or-more participants was 0.85 (0.75–0.97) (reference: non-participants). No significant association was observed between the number of days of medical treatment and participation in the program in either of the models.

The results of the negative binomial regression analyses for expenditures and the number of days of medical treatment in males and females are shown in Table 5. In males, the RR of medical care expenditures for the twice-or-more participants (reference: non-participants) was significantly lower in Model 1. In females, the RR of medical care expenditures for the twice-or-more participants (reference: non-participants) was significantly lower in Model 2. The RRs of expenditures and the number of days of medical treatment for the one-time participants (reference: non-participants) were significantly lower in Model 1 and Model 2. 

## 4. Discussion

The results of this longitudinal study involving Japanese workers showed that those who had participated twice or more in the short oral health promotion program in the workplace spent less on dental and periodontal care than non-participants after adjusting for sex and age. The results of the present study agree with those from the previous longitudinal studies that showed the association between participation in intensive workplace oral health promotion programs taking 20 min or longer per participant and reduction in dental and medical care expenditures [6,21]. Based on the results of the present study, it appears that our short workplace oral health promotion program may reduce dental and medical care expenditures.

There are other explanations for the association between participation in the program and dental and medical care expenditures. For example, program participants might have higher health literacy, including oral health literacy, than non-participants [22]. Subjects with high health literacy tend to brush their teeth more frequently, to be non-smokers, and to use fluoride toothpaste more often when compared to those with low health literacy. The results of this may be that the program participants might have less periodontal disease, dental caries, and systemic diseases, visit dentists less frequently, and spend less money for dental, periodontal, and medical treatments.

Results from the zero-inflated part of the zero-inflated negative binomial models for number of days of dental and periodontal treatment showed that all program participants, regardless of history of previous participation, were more likely to visit dentists for dental and periodontal treatment than non-participants. In particular, the ORs in the one-time participants were lower than those in the twice-or-more participants. Since the dentists in the program recommended that participants should routinely go for dental check-ups and scaling, these results show that the participants followed the advice given by these dentists. In particular, the one-time participants might have been advised not only to visit dentists routinely for dental check-ups, but also to visit dentists for treatment of dental diseases. The percentages of the one-time and twice-or-more participants having untreated dental caries and/or periodontal disease were 47.4% and 52.3%, respectively (data not shown). 

The twice-or-more participants spent significantly less on medical treatment than non-participants; however, no significant difference was found in the number of days of medical treatment between the twice-or-more participants and non-participants in the negative binomial regression models. These results showed that the twice-or-more participants had higher medical care expenditures per medical visit than non-participants, suggesting more serious diseases with higher cost. In fact, reduction of the RR value by adding diabetes mellitus in Model 2 of the negative binomial regression showed that the association between participation in the program and medical expenditures was confounded by having diabetes mellitus. Moreover, the oral health status of the non-participants in the workplace oral health promotion program was unknown; however, there might be an association between oral health status including an increased number of teeth present and low medical care expenditure in this study population, as suggested in previous studies [19]. The number of teeth present in the participants ranged from 17 to 32 (mean: 28.5, standard deviation: 1.8, median: 28, 25th percentile: 28, and 75th percentile: 30), which is more than the average in the Japanese population (percentages of subjects with 20 or more teeth in the 20 to 69-year-old population: 90.1%) [2].

Results of the zero-inflated negative binomial regression analyses and the negative binomial regression analyses after stratification by sex showed a difference in the strength of the association between participation in the program and expenditures and number of days of dental, periodontal, and medical treatment. For example, the associations between participation in the program twice or more and expenditures and the number of days of dental and periodontal treatment were significant in females, but not in males. The difference might be ascribed to the higher expenditures and number of days of dental and periodontal treatment in females than in males (data not shown). Because of the low number of one-time participants, both sexes were combined and adjustment for sex was performed in the main analyses. Additional research is required to explore sex differences in the association between participation in the program and expenditures and the number of days of dental and medical treatment. 

This study has some limitations. First, subjects who participated in the program before 2016 but not in 2017 were misclassified into the non-participant group. This might have resulted in underestimation of the association between participation in the program and expenditures and the number of days of treatment. Second, information on socioeconomic status, including educational status and income, was not obtained. However, the company is one of the most popular companies in the educational sector among job hunters. The percentage of university graduates and average annual income might be higher than in other companies in the same sector. The relatively high socioeconomic status of the study subjects might have resulted in the high participation rate in the program and the maintenance of good dental and periodontal health in the program participants. Third, no information was obtained regarding oral health status, including periodontal status and use of prosthetic devices in non-participants. Periodontal status in the participants in fiscal year 2017 was as follows: normal 56.2%, gingivitis 38.9%, and periodontitis 4.9%. Fourth, only dental and medical expenditures in the next year of program participation were evaluated in the present study. Additional research is required to evaluate long-term effects. 

## 5. Conclusions

This longitudinal study involving workers in Japan showed that participants in the short version of a workplace oral health promotion program visit dentists for dental and periodontal treatments. The twice-or-more participants spent less on dental, periodontal, and medical treatment than non-participants after adjusting for sex and age.

## Figures and Tables

**Figure 1 ijerph-19-03143-f001:**
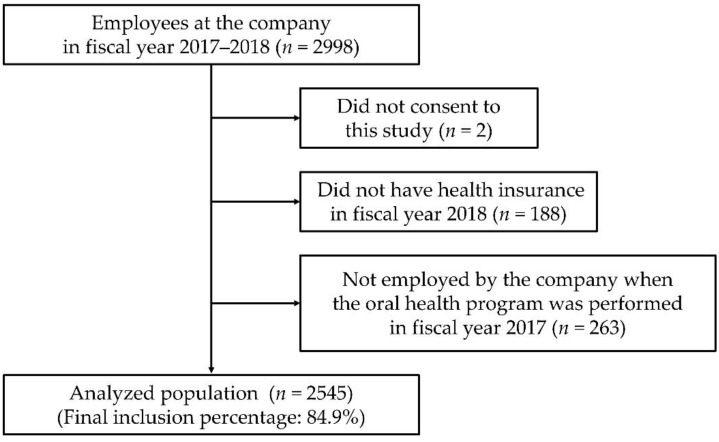
Flow chart of the participant selection process.

**Table 1 ijerph-19-03143-t001:** Comparison of subjects grouped by participation history in the short workplace oral health promotion program according to each variable.

		Total	Participation in Fiscal Year 2017 and Prior						*p* ^b^
			Never		Once		Twice or More		
		(*n* = 2545)	(*n* = 1258)		(*n* = 86)		(*n* = 1201)		
		*n*	*n*	%	*n*	%	*n*	%	
Sex	Male	1059	494	46.6	39	3.7	526	49.7	0.058
	Female	1486	764	51.4	47	3.2	675	45.4	
Diabetes mellitus	Without	2277	1120	49.2	79	3.5	1078	47.3	0.642
	With	268	138	51.5	7	2.6	123	45.9	
		%	Statistics		Statistics		Statistics		*p* ^c^
Age in fiscal year 2017 (y) ^d^	Median		41		30		39		<0.001
	25th percentile		34		24		32		
	75th percentile		47		40		46		
Dental care expenditures in fiscal year 2018 (Japanese yen ^a^)	Median		8950		12,480		8960		0.364
	25th percentile		0		0		0		
	75th percentile		30,690		37,988		28,560		
	Percentage of zeros	44.0%							
Periodontal care expenditures in fiscal year 2018 (Japanese yen)	Median		5295		8390		6950		0.732
	25th percentile		0		0		0		
	75th percentile		26,403		28,998		24,990		
	Percentage of zeros	48.7%							
Medical care expenditures in fiscal year 2018 (Japanese yen)	Median		30,530		31,035		37,060		0.083
	25th percentile		11,678		14,708		14,070		
	75th percentile		75,818		80,400		75,505		
	Percentage of zeros	4.4%							
Number of days of dental treatment in fiscal year 2018	Median		1.0		2.0		1.0		0.267
	25th percentile		0.0		0.0		0.0		
	75th percentile		4.0		5.0		4.0		
	Percentage of zeros	44.0%							
Number of days of periodontal treatment in fiscal year 2018	Median		1.0		1.0		1.0		0.703
	25th percentile		0.0		0.0		0.0		
	75th percentile		4.0		4.0		3.0		
	Percentage of zeros	48.7%							
Number of days of medical treatment in fiscal year 2018 ^e^	Median		5.0		5.0		6.0		0.013
	25th percentile		2.0		2.0		3.0		
	75th percentile		11.0		11.0		12.0		
	Percentage of zeros	4.5%							

^a^ 110 Japanese yen = 1 USD in 2018. ^b^ Chi-squared test. ^c^ Kruskal–Wallis test. ^d^ significant differences between each pair of the three groups by the Mann–Whitney *U* test with the Bonferroni correction. ^e^ significant difference between non-participants and twice-or-more participants by the Mann–Whitney *U* test with the Bonferroni correction.

**Table 2 ijerph-19-03143-t002:** Zero-inflated negative binomial regression models for expenditures and number of days of dental and periodontal treatment in all subjects.

Outcome Variable	Explanatory Variable		Zero-Inflated Part				Negative Binomial Part			
			OR ^a^	95% CI		*p*	RR ^b^	95% CI		*p*
				Low	High			Low	High	
Dental care expenditures in fiscal year 2018	Sex	Male	1.00	(reference)			1.00	(reference)		
		Female	0.59	0.40	0.88	0.010	0.84	0.83	0.85	<0.001
	Age		0.98	0.97	0.99	<0.001	1.01	0.98	1.04	0.666
	Participation in the program in the fiscal year 2017 and prior	Never	1.00	(reference)			1.00	(reference)		
		Once	0.47	0.35	0.62	<0.001	0.86	0.48	1.53	0.600
		Twice or more	0.82	0.64	1.05	0.115	0.84	0.83	0.85	<0.001
Periodontal care expenditures in fiscal year 2018	Sex	Male	1.00	(reference)			1.00	(reference)		
		Female	0.68	0.60	0.78	<0.001	0.85	0.83	0.88	<0.001
	Age		0.98	0.97	0.99	<0.001	1.01	0.98	1.04	0.689
	Participation in the program in the fiscal year 2017 and prior	Never	1.00	(reference)			1.00	(reference)		
		Once	0.48	0.35	0.67	<0.001	0.79	0.50	1.24	0.301
		Twice or more	0.79	0.67	0.92	0.003	0.85	0.83	0.87	<0.001
Number of days of dental treatment in fiscal year 2018	Sex	Male	1.00	(reference)			1.00	(reference)		
		Female	0.43	0.22	0.81	0.009	0.82	0.75	0.91	<0.001
	Age		0.98	0.97	0.99	<0.001	1.01	0.97	1.05	0.665
	Participation in the program in the fiscal year 2017 and prior	Never	1.00	(reference)			1.00	(reference)		
		Once	0.29	0.15	0.56	<0.001	0.83	0.35	1.98	0.668
		Twice or more	0.67	0.53	0.85	0.001	0.79	0.78	0.79	<0.001
Number of days of periodontal treatment in fiscal year 2018	Sex	Male	1.00	(reference)			1.00	(reference)		
		Female	0.53	0.41	0.67	<0.001	0.82	0.72	0.94	0.004
	Age		0.98	0.97	0.98	<0.001	1.01	0.97	1.05	0.695
	Participation in the program in the fiscal year 2017 and prior	Never	1.00	(reference)			1.00	(reference)		
		Once	0.27	0.14	0.53	<0.001	0.72	0.34	1.52	0.395
		Twice or more	0.65	0.55	0.76	<0.001	0.80	0.74	0.87	<0.001

^a^ Odds ratio. ^b^ Rate ratio.

**Table 3 ijerph-19-03143-t003:** Zero-inflated negative binomial regression models for expenditures and number of days of dental and periodontal treatment in males and females after adjusting for age.

Outcome Variable in Fiscal Year 2018	Participation in the Program in Fiscal Year 2017 and Prior (Reference: Never)	Zero-Inflated Part				Negative Binomial Part			
		OR ^a^	95% CI		*p*	RR ^b^	95% CI		*p*
			Low	High			Low	High	
Male									
Dental care	Once	0.25	0.24	0.26	<0.001	0.75	0.36	1.56	0.445
expenditures	Twice or more	0.73	0.51	1.04	0.085	0.85	0.68	1.06	0.153
Periodontal care	Once	0.22	0.17	0.29	<0.001	0.68	0.40	1.17	0.166
expenditures	Twice or more	0.72	0.52	1.00	0.048	0.87	0.66	1.16	0.338
Number of days of	Once	0.15	0.09	0.23	<0.001	0.76	0.22	2.62	0.669
dental treatment	Twice or more	0.65	0.40	1.05	0.076	0.81	0.58	1.14	0.226
Number of days of	Once	0.07	0.05	0.12	<0.001	0.68	0.22	2.09	0.500
periodontal treatment	Twice or more	0.63	0.38	1.06	0.080	0.83	0.53	1.32	0.436
Female									
Dental care	Once	0.56	0.16	1.92	0.355	0.82	0.35	1.90	0.644
Dexpenditures	Twice or more	0.80	0.54	1.18	0.255	0.79	0.76	0.82	<0.001
Periodontal care	Once	0.65	0.14	2.90	0.570	0.77	0.35	1.71	0.521
expenditures	Twice or more	0.75	0.58	0.97	0.031	0.79	0.77	0.81	<0.001
Number of days of	Once	0.43	0.05	3.95	0.457	0.74	0.23	2.39	0.619
dental treatment	Twice or more	0.56	0.37	0.86	0.008	0.72	0.69	0.75	<0.001
Number of days of	Once	0.53	0.05	5.19	0.588	0.64	0.21	1.93	0.427
periodontal treatment	Twice or more	0.54	0.43	0.69	<0.001	0.72	0.71	0.73	<0.001

^a^ Odds ratio; ^b^ Rate ratio.

**Table 4 ijerph-19-03143-t004:** Negative binomial regression models for expenditures and number of days of medical treatment in all subjects.

Outcome Variable	Explanatory Variable		Model 1				Model 2			
			RR ^a^	95% CI		*p*	RR ^a^	95% CI		*p*
				Low	High			Low	High	
Medical care expenditures in fiscal year 2018	Sex	Male	1.00	(reference)			1.00	(reference)		
		Female	1.00	0.63	1.57	0.989	1.16	0.56	2.38	0.691
	Age (y)		1.02	0.96	1.09	0.518	1.01	0.97	1.07	0.575
	Diabetes mellitus	Without	1.00	(reference)			1.00	(reference)		
		With					4.07	0.13	127.49	0.425
	Participation in the program in the fiscal year 2017 and prior	Never	1.00	(reference)			1.00	(reference)		
		Once	0.77	0.36	1.67	0.511	0.89	0.33	2.37	0.814
		Twice or more	0.81	0.77	0.85	<0.001	0.85	0.75	0.97	0.014
Number of days of medical treatment in fiscal year 2018	Sex	Male	1.00	(reference)			1.00	(reference)		
		Female	1.17	0.63	2.16	0.620	1.23	0.61	2.47	0.568
	Age (y)		1.01	0.98	1.04	0.625	1.00	0.98	1.02	0.839
	Diabetes mellitus	Without	1.00	(reference)			1.00	(reference)		
		With					2.05	0.31	13.49	0.453
	Participation in the program in the fiscal year 2017 and prior	Never	1.00	(reference)			1.00	(reference)		
		Once	0.84	0.50	1.40	0.493	0.83	0.52	1.32	0.429
		Twice or more	1.00	0.74	1.35	0.990	1.00	0.74	1.33	0.975

^a^ Rate ratio.

**Table 5 ijerph-19-03143-t005:** Negative binomial regression models for expenditures and number of days of medical treatment in males and females.

Outcome Variable in Fiscal Year 2018	Participation in the Program in Fiscal Year 2017 and Prior (Reference: Never)	Model 1				Model 2			
		RR ^a^	95% CI		*p*	RR ^a^	95% CI		*p*
			Low	High			Low	High	
Male									
Medical care	Once	0.37	0.33	0.42	<0.001	0.46	0.31	0.69	<0.001
expenditures	Twice or more	0.65	0.60	0.70	<0.001	0.90	0.53	1.52	0.688
Number of days of medical treatment	Once	0.62	0.41	0.93	0.022	0.59	0.44	0.80	0.001
	Twice or more	0.90	0.67	1.20	0.461	0.92	0.67	1.27	0.601
Female		1.02	0.95	1.09	0.574	1.01	0.96	1.07	0.640
Medical care	Once	0.96	0.20	4.53	0.960	1.06	0.20	5.56	0.947
expenditures	Twice or more	0.88	0.65	1.19	0.407	0.77	0.75	0.78	<0.001
Number of days of medical treatment	Once	0.85	0.38	1.90	0.696	0.87	0.39	1.98	0.749
	Twice or more	1.02	0.67	1.57	0.922	1.00	0.69	1.45	0.997

^a^ Rate ratio. Model 1: adjusting for age. Model 2: adjusting for age and diabetes mellitus.

## Data Availability

The data presented in this study are available on request from the corresponding author. The data are not publicly available due to ethical restrictions.

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
