# Peer review of "Association between Participation in the Short Version of a Workplace Oral Health Promotion Program and Medical and Dental Care Expenditures in Japanese Workers: A Longitudinal Study"

_ijerph, 2022, doi:10.3390/ijerph19053143_

Round 1

Reviewer 1 Report

Who are the participants?? It is not clear about their background. 

It is not clear about the educational status and percapita income both of which might have a potential role. Also, 188 subjects didn't have health insurance within the organisation. Is it because they could not afford the insurance because their income levels are low or they have so bad oral health that they were not insured? Can the results be biased because of this?

It is necessary to put a flow diagram to make readers understand the study process. 

More details of the program needs to mentioned. Who conducted the program, mode and language of administration, etc..  

Introduction: We have already confirmed the effectiveness of 61 the instructional portion of the program, such as through improvement of periodontal disease [12]. - What do you mean by this? The reference quoted is your earlier study?

Table 1 showed that there was no significant difference with various parameters versus non-participants, one time or twice or more participants. Hence it is not clear why the authors performed zero-inflated models and NB regressions

Author Response

Response to Reviewer 1:

Who are the participants?? It is not clear about their background.

It is not clear about the educational status and percapita income both of which might have a potential role. Also, 188 subjects didn't have health insurance within the organisation. Is it because they could not afford the insurance because their income levels are low or they have so bad oral health that they were not insured? Can the results be biased because of this?

Our response: Thank you for your comments. As the reviewer pointed out, educational status and income might have a potential role. The lack of this information is a limitation of the present study; therefore, we have added an explanation (Lines 319-325).

The 188 subjects did not have health insurance within the organization but they had health insurance within another organization because Japan has a universal health care system. The reason for not having health insurance within the organization was not that their income levels were low or that they had bad oral health. Instead, the main reason for not having health insurance within the organization might have been retirement before March 2019. The service the person received was the same regardless of the insurance the person had. Therefore, the results were not biased because of that.

It is necessary to put a flow diagram to make readers understand the study process.

Our response: We have added a flow diagram (Figure 1).

More details of the program need to mentioned. Who conducted the program, mode and language of administration, etc.. 

Our response: We have added the information requested (Lines 91-94).

Introduction: We have already confirmed the effectiveness of 61 the instructional portion of the program, such as through improvement of periodontal disease [12]. - What do you mean by this? The reference quoted is your earlier study?

Our response: Yes, the reference quoted is our earlier study. We have added that information (Line 64).

Table 1 showed that there was no significant difference with various parameters versus non-participants, one time or twice or more participants. Hence it is not clear why the authors performed zero-inflated models and NB regressions

Our response: We performed zero-inflated models and NB regressions, because sex and age might have confounded the association between participation in the program and medical and dental expenditures. In fact, previous studies have shown that medical and dental expenditures vary by sex and age. We have added that explanation, citing the paper (Lines 151-152).

Reviewer 2 Report

The study has many limitations that have not been described.

  1. Not clearly explained why diabetes was chosen, and no other underlying conditions were correlated in the study.
  2. Authors missed looking into the use of prosthetic devices. Removable vs. Fixed. Implants.
  3. Preexisting periodontal disease was never taken into consideration. Why? It should be a discussion about all these missing factors.
  4. No discussion about an extended period of study and possible effects
  5. Never discuss eduction level of participants and impact on following instructions

Exploring and addressing these limitations will make the paper stronger

Author Response

The study has many limitations that have not been described.

Not clearly explained why diabetes was chosen, and no other underlying conditions were correlated in the study.

Our response: Thank you for your comments. Diabetes mellitus was chosen because it is associated with periodontal disease and may have confounded the association between periodontal tissue improvement through participation and medical care costs in this study. Unfortunately, we could not obtain information about other underlying conditions. We have added an explanation of this to the text (Lines 160-161).

Authors missed looking into the use of prosthetic devices. Removable vs. Fixed. Implants.

Our response: Unfortunately, we could not obtain the information regarding the use of prosthetic devices. We have added this as a limitation (Lines 325-327).

Preexisting periodontal disease was never taken into consideration. Why? It should be a discussion about all these missing factors.

Our response: Because we could not obtain the information about periodontal status of the non-participants, we could not evaluate the preexisting periodontal disease in the present study. We have added this as a limitation, and we have added the prevalence of periodontal disease in the participants (Lines 325-328).

No discussion about an extended period of study and possible effects.

Our response: The short study period is a limitation of the present study. We have added it as a limitation (Lines 328-330).

Never discuss education level of participants and impact on following instructions.

Our response: We have added the lack of this information as a limitation (Lines 319-325).

Reviewer 3 Report

We read with great interest the manuscript with title: “Association between participation in the short version of a workplace oral health promotion program and medical and dental care expenditures in Japanese workers: A longitudinal study” aiming to to evaluate the short version of an oral health promotion program in the workplace from the viewpoint of dental and medical care expenditures.

The study is of interest; however, several criticisms have been found and need to be addressed before resubmission.

Introduction

  1. Line 29-30 please specify the time frame for the improvement of oral health in children
  2. Line 73-81 please change the order of sentences: the sentence clarifying the aim of the study should be the last of the Introduction

M&M

  1. The characteristics of the participants should be moved to the Result section. In the M&M section authors should state inclusion /exclusion criteria, but not the characteristics of the sample. (Number, age, gender, etc.)
  2. Please state if the rapid measurement of lactate dehydrogenase in saliva required specific conditions (abstaining from drinking, eating, and if yes please state the duration of these conditions prior examination)
  3. “While waiting for test results, examination of the oral cavity, including for numbers of teeth present and decayed teeth, was performed by a dentist. “Did authors perform the DMFT index?
  4. “When the participants had dental problems and wanted consultation from the dentist, the time spent tended to be longer”. Please state how longer the time was, if subjects needed more consultation.
  5. “The participation rate in the short oral health program in fiscal year 2017 was 50.6%, 111 with 3.4% being first-time participants and 47.2% being those participating for at least a 112 second time (Table 1) “. This must be moved to Result section.

Results

  1. Please present data with a gender differentiated approach, data for female and male for each studied condition. Please add another Table to present these data.
  2. Please start the Results section with a description of the enrolled sample.

Author Response

Introduction

Line 29-30 please specify the time frame for the improvement of oral health in children

Our response: Thank you for your comments. We have added the information (Line 30).

Line 73-81 please change the order of sentences: the sentence clarifying the aim of the study should be the last of the Introduction

Our response: We have revised the text accordingly (Lines 78-81).

M&M

The characteristics of the participants should be moved to the Result section. In the M&M section authors should state inclusion /exclusion criteria, but not the characteristics of the sample. (Number, age, gender, etc.)

Our response: We have revised the text as suggested (Lines 86-90 and 170-174).

Please state if the rapid measurement of lactate dehydrogenase in saliva required specific conditions (abstaining from drinking, eating, and if yes please state the duration of these conditions prior examination)

Our response: The rapid measurement of lactate dehydrogenase in saliva did not require specific conditions. We have added this information (Lines 104-105).

“While waiting for test results, examination of the oral cavity, including for numbers of teeth present and decayed teeth, was performed by a dentist. “Did authors perform the DMFT index?

Our response: We checked only the number of D (decayed) teeth to let the participant know their dental health status and to recommend a visit to a dentist if the participant had decayed teeth. We have added this information (Lines 107-109).

“When the participants had dental problems and wanted consultation from the dentist, the time spent tended to be longer”. Please state how longer the time was, if subjects needed more consultation.

Our response: We have added an explanation as suggested (Lines 115-116).

“The participation rate in the short oral health program in fiscal year 2017 was 50.6%, 111 with 3.4% being first-time participants and 47.2% being those participating for at least a 112 second time (Table 1) “. This must be moved to Result section.

Our response: We have revised the text as suggested (Lines 179-181).

Results

Please present data with a gender differentiated approach, data for female and male for each studied condition. Please add another Table to present these data.

Our response: We have added the analyses according to the reviewer’s comment (Lines 164-166, 222-228, 245-252 and 305-316, and Tables 3 and 5). However, the conclusions were drawn from the analyses using all subjects combined, including both sexes and then adjusted for sex, because the number of one-time participants was low, which might make the statistical models unstable.

Please start the Results section with a description of the enrolled sample.

Our response: We have revised the text accordingly (Lines 170-174).

Round 2

Reviewer 1 Report

All my comments were addressed.